# Cognitive and affective empathy in binge drinking during late adolescence

Brooke A. Lester[1], Nita Bislimi[2], Claus Lamm[2], Ekaterina Pronizius[2‡*]

**1** School of Neuroscience, Virginia Polytechnic Institute and State University, United States of America,
**2** Department of Cognition, Emotion, and Methods, Faculty of Psychology, University of Vienna, Austria

‡ This author is the senior and the corresponding author of this work.
* ekaterina.pronizius@univie.ac.at

## Abstract

Empathy–the ability to understand and share the feelings of others–has been identified as a potential correlate of binge drinking behaviors in adolescence. Our preregistered study aimed to investigate the relationship between cognitive and affective empathy and alcohol consumption in late adolescence. Building on prior research, we predicted a dissociation of cognitive and affective empathy related to alcohol consumption. Specifically, we hypothesized that whereas a negative association between affective empathy and alcohol consumption would emerge, no significant association would be observed for cognitive empathy. To test these hypotheses, we surveyed a convenience sample of U.S. college students (*N* = 116; ages 18–23; 73.28% women). The online survey comprised questions assessing drinking behavior (categorized as non-binge drinking, binge drinking, and extreme binge drinking), engagement in hazardous drinking behavior (continuous measure), and self-reported empathic traits. We found that binge drinking categories and hazardous drinking scores were closely aligned, suggesting that the two measures capture similar aspects of drinking behavior. Men reported more problematic drinking patterns than women, while women scored higher on affective empathy. Age was positively related to hazardous drinking, with participants at the upper end of the age range reporting riskier alcohol use. Contrary to our preregistered hypotheses, we found no significant relationship between affective empathy and engagement in binge drinking or hazardous drinking among late adolescents. Cognitive empathy, as predicted, was not related to the extremity of alcohol consumption. Additionally, we observed a striking pattern in our data: 20.69% of our sample indicated extreme binge drinking (EBD), which is higher than the 12% reported by a U.S. national survey conducted in 2020. While the relationship between drinking behavior and empathy may be more nuanced than expected, the observed disparity in EBD behavior emphasizes the importance of addressing its prevalence within U.S. college communities and implementing proactive measures to promote responsible alcohol consumption during late adolescence.

**Data availability statement:** All data files necessary for alternative analyses and for extensions and replications are available from the OSF database (DOI 10.17605/OSF.IO/V2KCF; link: https://osf.io/v2kcf/).

**Funding:** The author(s) received no specific funding for this work.

**Competing interests:** The authors have declared that no competing interests exist.

## Introduction

Harmful alcohol use during late adolescence represents a significant public health concern due to its high prevalence, associated risks, and enduring consequences for both health and social functioning [(SAMHSA, 1,see 2, 3 for reviews)]. While binge drinking (BD) has been the focus of considerable research, far less is known about high-intensity/extreme binge drinking (EBD), despite emerging evidence linking it to particularly severe outcomes [(see 4, 5 for reviews)]. Late adolescence, a period of socio-emotional and social brain maturation, is a developmental stage in which empathy may be critical to understanding why some youth escalate from recreational to hazardous drinking, including BD and EBD.

BD is defined as a pattern of alcohol consumption that raises blood alcohol concentration (BAC) to 0.08% (0.08 grams per deciliter) or higher; for a typical adult, this level is generally reached after consuming five or more standard drinks for men, or four or more standard drinks for women, within approximately two hours [6,see 7 for review, but see 8 for a critique]. According to the U.S. Substance Abuse and Mental Health Services Administration's 2021 National Survey on Drug Use and Health, 29.2% of young adults aged 18–25 (about 9.8 million) reported past-month BD, higher than among adults aged 26 or older and adolescents aged 12–17 (SAMHSA, [1]). An emerging behavioral pattern of concern within the United States is high-intensity drinking [9,see 5, 10, 4 for reviews], or extreme BD (EBD), as it will be referred to here. EBD is the consumption of alcohol at two or more times the gender-specific levels of BD (10 or more standard drinks at one time for men or 8 or more standard drinks at one time for women) [5]. Compared with non-binge drinkers (non-BD), individuals consuming between two and three times the gender-specific BD threshold had approximately ~70-fold higher odds of an alcohol-related emergency department visit, and those at three times or more had ~93-fold higher odds [4]. In 2020, the prevalence of EBD was 12% among both college students and their non-college peers [11].

Late adolescence (ages 18–24) represents a developmental window of heightened vulnerability to such behaviors. Although the World Health Organization defines adolescence as ages 10–19 [WHO, 12], extensions into the early twenties are supported by both shifts in societal norms and advances in knowledge about brain development [see 13–15 for reviews]. In particular, studies highlight that the maturation of the "social brain" continues well into the early twenties, with developmental changes extending up to approximately age 24 [see 13–15 for reviews]. This prolonged period of growth coincides with heightened sensitivity to social (peer) influences. One important manifestation of this is social facilitation, defined as the tendency to increase or modify drinking behaviors in the presence of peers, which has been identified as a central factor shaping adolescent alcohol use [16, 17,see 13, 7 for reviews].

Across studies that define "extreme" drinking differently—some by very high single-episode quantities (e.g., 10+/8+drinks or ≥2–3×binge thresholds), others by frequent monthly binges—a consistent pattern emerges: motives differ between standard binge drinkers and extreme binge drinkers. Research with college samples has found that students' drinking motives, specifically social (drinking to enhance

interactions), enhancement (drinking to increase positive affect or excitement), and coping (drinking to reduce stress or negative emotions), distinguish extreme drinkers from both their binge and non-binge peers. Moreover, changes in these motives over time align with transitions into or out of EBD behavior [18]. Within binge drinkers, a "hazardous" subgroup (frequent bingers) shows the highest coping and conformity motives (and higher social/enhancement), alongside more frequent bingeing and more problems than other binge-drinker clusters [19]. A study using a national cohort–based sample of young adult drinkers found that while BD and EBD have overlapping risk profiles, high-intensity drinkers reported higher perceived heavy-drinking norms and stronger social/enhancement motives [20]. Given rising hazardous alcohol use among young adult drinkers, these differentiated motive profiles underscore the need to examine drinking intensities beyond the standard binge definition, along with their associated risk factors (in line with [5]).

Empathy, the ability to share or resonate with the affective states of others [21,22], has been suggested as a potential correlate of BD tendencies across different age groups and populations (adolescents: [23, 24], adults: [25], for a meta-analysis in clinical and non-clinical samples, [26]). Crucially, empathy undergoes refinement throughout late adolescence [27,28]. As noted previously, social facilitation has been identified as a central factor shaping adolescent alcohol use [17, 16, see 13, 7 for reviews]. In this context of heightened sensitivity to peer influence, empathy may therefore amplify drinking dynamics by further increasing adolescents' responsiveness to peers' emotions [see 29 for a meta-analysis] and by enhancing the salience and impact of social contexts [24, see 14 for review].

Although conceptualizations of empathy vary greatly across the literature [30, see 31 for review], it is most commonly subdivided into two broad components: cognitive empathy and affective empathy [32, but see 33 for critical review]. These components constitute interacting systems, ultimately enabling individuals to understand and resonate with the feelings and thoughts of others [32,34]. Cognitive empathy refers to controlled, deliberate processes that involve understanding others' perspectives and imagining their experiences, whereas affective empathy involves automatic, emotional responses that allow one to share or resonate with others' feelings [i.e., affect sharing, 34, 32, but see 33 for critical review].

Previous research has shown that alcohol intake is not associated with a general reduction in empathy, but rather with a dissociative pattern across its subcomponents. For instance, some studies have reported impaired affective empathy alongside preserved cognitive empathy in adults who misuse alcohol [35] as well as in adolescent binge drinkers [24]. In contrast, a meta-analysis found the opposite pattern in adults with alcohol use disorder (AUD): while empathy impairments were associated with AUD, only cognitive empathy was affected when its subcomponents were considered [26]. Another meta-analysis reported that both cognitive and affective empathy were negatively associated with concurrent substance use, and that increases in affective empathy over time predicted decreases in substance use [36].

Despite these broad associations evidenced in prior research, the exact nature of the relationship between alcohol consumption and empathy, whether causal, consequential, or bidirectional, remains unknown. Massey et al. [37] advanced a theoretical model suggesting that variation along the empathic capacity spectrum may influence the progression of substance use. They hypothesized that reduced empathy diminishes sensitivity to punishment and social disapproval, thereby enabling escalation of use. Conversely, higher empathy may function as a protective factor by enhancing responsiveness to others' distress, thereby discouraging continued use. Consistent with this, Martinotti et al. [38] found reduced empathy among alcohol-dependent patients and proposed that diminished empathy may prompt some individuals to use alcohol as a compensatory strategy. A longitudinal study of adolescents admitted to substance use treatment programs across six large U.S. cities found that higher cognitive empathy predicted stronger responsiveness to the social consequences of substance use. In turn, responsiveness to these consequences at treatment completion was associated with a steeper decline in substance use at follow-up [39]. Therefore, empathy may not only shape how individuals respond to social and disciplinary cues but also influence whether they initiate, escalate, or reduce alcohol use.

Given the outlined importance of distinguishing between extremes of drinking behavior and the distinct motivations underlying varying levels of alcohol consumption, the question remains whether empathy is differentially associated with

EBD versus BD. The present study addresses this by investigating the relationship between cognitive and affective empathy and two extreme levels of alcohol consumption (BD and EBD) within a sample of late adolescents. Specifically, we examine whether individuals engaged in BD and EBD exhibit differences in their empathic traits compared to non-binge drinkers (non-BD). In addition, we assess hazardous drinking continuously to evaluate its convergence with the categorical grouping (non-BD, BD, EBD). In line with prior research [24,35], we expect a dissociation between cognitive and affective empathy in BD and EBD participants. Specifically, we predict a negative relationship between the level of alcohol consumption and affective empathy, while cognitive empathy is not expected to show significant associations.

## Materials and methods

### Participants

A convenience sample was recruited via social media, listservs, and word of mouth, targeting students enrolled at two nearby U.S. universities. Participants were recruited between July 11th and August 1st, 2023. The preregistered exclusion criteria were: (a) not attending a U.S. university (self-reported), (b) being outside the age range of 18–24 years, (c) not identifying as either a woman or a man (due to the binary gender-specific BD standards applied in the analysis), and (d) incomplete survey responses. The final sample comprised N = 116 participants (women = 73.28%, age range 18–23, $M_{age}$ = 19.93, $SD_{age}$ = 1.24).

### Procedure

The study was conducted entirely online via the online survey platform Qualtrics, accessible on participants' personal computers or mobile devices. After providing informed consent, participants reported demographic information including age, gender, and race/ethnicity. Recruitment targeted students at the University of North Carolina at Chapel Hill and North Carolina State University; however, institutional affiliation was not collected to protect participant privacy. Next, the participants were asked questions regarding their drinking behaviors, followed by a questionnaire measuring affective and cognitive empathy. Upon completion, participants received a full debriefing via a document implemented within Qualtrics. The study took approximately 20 minutes to complete.

### Measures

**Demographic variables.** Participants indicated their age using a categorical scale ranging from 18–24 years in one-year increments, reported their gender (1 = man, 2 = woman), and selected their race/ethnicity from a predefined list (1 = Native American/Alaska Native, 2 = Asian/Pacific Islander, 3 = Black or African American, 4 = Hispanic, 5 = White/Caucasian, 6 = Multiple ethnicities/Other, 7 = Prefer not to say).

### Patterns of alcohol consumption

Patterns of alcohol consumption were evaluated using two self-report measures. In accordance with the definition of BD provided by the National Institute on Alcohol Abuse and Alcoholism ([6],also, [7]), BD was defined as five or more drinks in two hours for men and four or more drinks in two hours for women (NIAAA, [6]). EBD was defined as consuming ten or more drinks in one sitting for men and eight or more drinks in one sitting for women [5]. To be classified into the BD or EBD group, participants were required to have engaged in the respective behavior at least twice within the past six months (in line with [7], for review). The non-BD group consisted of those who did not meet the criteria for either BD or EBD. Individuals who met the EBD criteria were assigned to the EBD group and excluded from the BD group. Therefore, the three drinking groups (non-BD, BD, and EBD) were constructed to be mutually exclusive to ensure independence.

The second measure used for the continuous variable of hazardous alcohol consumption was the Alcohol Use Disorders Identification Test – Consumption (AUDIT-C) [40], which assesses alcohol consumption frequency, intensity, and

the frequency of consuming more than six drinks on a single occasion. The AUDIT-C is a reliable measure for assessing problematic alcohol consumption in adolescents [41] and college students [42], and has been validated in both online and in-clinic modalities [43].

## Empathy questionnaire

The Questionnaire of Cognitive and Affective Empathy (QCAE) is a measure of trait empathy comprising two subscales: cognitive empathy (19 items) and affective empathy (12 items) [44]. Participants rated the extent to which each statement described them on a 4-point scale ranging from 1 (strongly disagree) to 4 (strongly agree). Higher scores on the respective subscales indicate higher levels of empathic ability. The QCAE [44] has been found to have both good reliability and validity in both self-report surveys and clinical samples [44,45]. In the current sample it demonstrated adequate internal consistency (cognitive empathy α = 0.82, affective empathy scale α = 0.77).

## Attention check

Given the online format of the study, an attention check item was implemented within the QCAE [44] to ensure data quality by verifying that participants read the items carefully. The item read: "*Please select 'strongly disagree' to demonstrate that you are paying attention to this question*".

## Preregistration

The study design and data analysis were preregistered: https://aspredicted.org/Q89_ZSN.

## Inclusivity in global research

The study was conducted in accordance with the Declaration of Helsinki (1964 and its later amendments) and approved by the Ethical Board of the University of Vienna, Faculty of Psychology (EK reference number 00577, amendment 16 to the project 00412). Prior to study participation, participants were presented with an information sheet outlining the study's purpose, procedures, and their rights as research participants. Informed consent was obtained electronically, with participants indicating agreement by selecting 'I agree.' This confirmed their understanding of the study, that their participation was voluntary and uncompensated, and their right to withdraw at any time without penalty. All responses were collected in an anonymized format.

Additional details on the ethical, cultural and scientific considerations related to inclusivity in global research are provided in the Supporting Information (S1 File).

## Data analysis

Analyses were conducted using JASP [46], (ver. 0.18.3) and R [47], (ver. 4.4.0). The data analysis followed the preregistered plan, and the full dataset is available online [48].

First, we conducted correlation analyses to examine the relationships among drinking group (three levels: non-BD, BD, and EBD), hazardous drinking (AUDIT-C; [40]), cognitive and affective empathy (QCAE; [44]), and demographic variables (age, gender).

Next, we conducted analyses with drinking group as a factor (non-BD, BD, and EBD), testing the effects of cognitive and affective empathy using a multinomial regression analysis, followed by a one-way analysis of variance (ANOVA). We then used a linear regression model to assess the effects of empathy on hazardous alcohol consumption (AUDIT-C).

For the multinomial regression analysis, the dependent variable was the drinking group (non-BD, BD, EBD) with the non-BD group serving as the reference group. The independent variables were cognitive empathy, affective empathy, and the demographic variables (age, gender; see Equation 1).

$$Group\ (non-BD,\ BD,\ EBD)\ \sim\ cognitive\ empathy\ +\ affective\ empathy\ +\ age\ +\ gender \tag{1}$$

We then conducted ANOVAs to test whether there were differences in cognitive and affective empathy across the three drinking groups.

To examine the relationship between empathy and hazardous drinking, we conducted a linear regression analysis with cognitive empathy, affective empathy, and control variables (age, gender) as the independent variables and the AUDIT-C scores as the dependent variable (see Equation 2).

$$AUDIT\text{-}C\ score\ \sim\ cognitive\ empathy\ +\ affective\ empathy\ +\ age\ +\ gender \tag{2}$$

For all analyses performed, the significance level was set at $p < .05$.

## Results

### Sample characteristics

Out of 128 participants, six participants were excluded for failing the attention check. In addition to preregistered exclusions, we applied two data-quality/outlier procedures not specified a priori. First, to address potential careless responding, we calculated Mahalanobis distance for 31 of our QCAE items using the R package "careless" [49]. Three participants were excluded as multivariate outliers, with Mahalanobis distances exceeding the $\chi^2$ cutoff at the 99% confidence level ($p < .01$). Second, three additional participants were excluded because their empathy scores (cognitive or affective, [44]) or AUDIT-C scores [40] deviated by more than ±2.5 SD from the group mean, indicating extreme cases.

The final sample comprised $N = 116$ participants (73.28% women), aged 18-23 years ($M_{age} = 19.93$, $SD_{age} = 1.24$). An a priori power analysis conducted using G*power [50] indicated that a minimum sample of 100 participants would provide 80% power to detect a medium-to-large effect size (f = .32) in a one-way ANOVA at α = .05 with three groups. Most participants identified as White/Caucasian (69.83%). Of the total sample, 46.55% were in the non-BD group (n = 54), 32.76% of participants were in the BD group (n = 38), and 20.69% of participants were in the EBD group (n = 24). The mean AUDIT-C score across participants was 3.61 (SD = 2.63). See Table 1 for detailed sample characteristics.

As shown in Table 2, men ($n = 31$) had a mean AUDIT-C score of 4.84 ($SD = 2.83$), whereas women ($n = 85$) had a mean score of 3.17 ($SD = 2.41$). A score of 3 or more for women and 4 or more for men is considered hazardous ([51], but see [52] for calibrated cutoffs). Thus, on average, both men and women in this sample exceeded the gender-specific thresholds for hazardous drinking.

### Statistical analyses

To obtain a complete overview of the relationships among the key study variables, we computed Spearman's rho correlations (appropriate given the ordinal nature of some variables) between drinking group (with three levels, non-BD = 0, BD = 1, EBD = 2), AUDIT-C scores, affective and cognitive empathy, age (18-23), and gender (coded: 1 = man, 2 = woman). The results are presented in Table 3.

A strong positive correlation between drinking group (non-BD, BD, and EBD) and AUDIT-C scores was found ($r_s = .80$, $p < .001$), indicating that these two measures (group categorization and AUDIT-C scores) align in their assessment of drinking behavior. Gender (coded 1 = man, 2 = woman) was negatively correlated with AUDIT-C scores ($r_s = -.27$, $p = .004$) and with the drinking group ($r_s = -.22$, $p = .019$), indicating more problematic overall drinking behavior in men. Gender was positively correlated with affective empathy ($r_s = .40$, $p < .001$), suggesting that women tend to show higher affective empathy scores. Finally, age significantly correlated with AUDIT-C scores ($r_s = .23$, $p = .01$), such that higher age was associated with higher hazardous drinking scores. Fig 1 presents mean scores on the AUDIT-C, QCAE cognitive empathy, and QCAE affective empathy, separated by drinking group.

**Table 1. Descriptive Statistics for the total sample and drinking groups.**

| Characteristics | Total Sample (N = 116) | Non-BD (n = 54) | BD (n = 38) | EBD (n = 24) |
|---|---|---|---|---|
| **Age** M (SD), Range | 19.93 (1.24),18–23 | 19.72 (1.32),18–23 | 20 (1.09),18–22 | 20.29 (1.20), 18–23 |
| **Gender** % (n) | | | | |
| Men | 26.72% (31) | 18.52% (10) | 26.32% (10) | 45.83% (11) |
| Women | 73.28% (85) | 81.48% (44) | 73.68% (28) | 54.17% (13) |
| **Ethnicity** % (n) | | | | |
| White | 69.83% (81) | 66.67% (36) | 63.16% (24) | 87.50% (21) |
| Hispanic | 9.48% (11) | 9.26% (5) | 13.16% (5) | 4.17% (1) |
| Asian | 8.62% (10) | 11.11% (6) | 7.90% (3) | 4.17% (1) |
| African American | 6.03% (7) | 5.56% (3) | 10.53% (4) | 0 |
| Multiple/Other | 6.03% (7) | 7.41% (4) | 5.26% (2) | 4.17% (1) |
| **AUDIT-C Score** M (SD), Range | 3.61 (2.63), 0–10 | 1.63 (1.47), 0–5 | 4.21 (1.47), 1–7 | 7.13 (1.83), 3–10 |
| **QCAE Cognitive Empathy** Score M (SD), Range | 61.35 (6.66), 43–76 | 61.17 (6.54), 46–75 | 62.08 (6.10), 47–73 | 60.63 (7.86), 43–76 |
| **QCAE Affective Empathy** Score M (SD), Range | 36.41 (5.22), 23–47 | 36.48 (5.05), 24–47 | 36.84 (5.14), 24–45 | 35.54 (5.82), 23–45 |

*Note.* M = mean, SD = standard deviation. N = 116. Non-BD = Non-binge drinking; BD = binge drinking; EBD = extreme binge drinking, AUDIT-C [40], QCAE [44]. Percentages may not total 100 due to rounding.

**Table 2. Descriptive statistics of the sample by gender.**

| | Men (n = 31) | | | Women (n = 85) | | |
|---|---|---|---|---|---|---|
| | M | SD | Range | M | SD | Range |
| Age (years) | 20.16 | 1.24 | 18–23 | 19.85 | 1.23 | 18–23 |
| AUDIT-C Score | 4.84 | 2.83 | 0–10 | 3.17 | 2.41 | 0–9 |
| QCAE Cognitive Empathy Score* | 59.97 | 5.24 | 50–73 | 61.86 | 7.07 | 43–76 |
| QCAE Affective Empathy Score* | 32.97 | 4.89 | 23–44 | 37.66 | 4.78 | 24–47 |

M = Mean, SD = standard deviation (SD), and range of values for age, AUDIT-C [40], and QCAE subscales [44] are shown, split by gender. Total N = 116.

*The QCAE values of our sample are comparable to those reported in the original validation study [44]: cognitive empathy, females M = 59.42 (SE = 0.30), males M = 56.12 (SE = 0.50); affective empathy, females M = 36.76 (SE = 0.20), and males M = 32.27 (SE = 0.30).

**Table 3. Correlation matrix.**

| | Age | Gender | Drinking Group | AUDIT-C [40] | QCAE Cognitive Empathy [44] | QCAE Affective Empathy [44] |
|---|---|---|---|---|---|---|
| **Age** | – | | | | | |
| **Gender** | −0.13 [-0.30, 0.06] | – | | | | |
| **Drinking Group** | 0.17 [-0.01, 0.35] | **−0.22*** **[-0.39, -0.04]** | – | | | |
| **AUDIT-C** | **0.23*** **[0.05, 0.40]** | **−0.27**** **[-0.43, -0.09]** | **0.80**** **[0.72, 0.86]** | – | | |
| **QCAE Cognitive Empathy** | −0.08 [-0.26, 0.11] | 0.16 [-0.02, 0.33] | 0.01 [-0.18, 0.19] | 0.02 [-0.17, 0.20] | – | |
| **QCAE Affective Empathy** | −0.16 [-0.33, 0.03] | **0.40**** **[0.23, 0.54]** | −0.02 [-0.20, 0.17] | 0.01 [-0.18, 0.19] | 0.16 [-0.02, 0.33] | – |

Spearman's rho correlation coefficients are presented. Values in square brackets indicate the 95% confidence interval for each correlation. The confidence interval is a plausible range of population correlations that could have caused the sample correlation [53].

* = p < .05, ** = p < .01, *** = p < .001

 

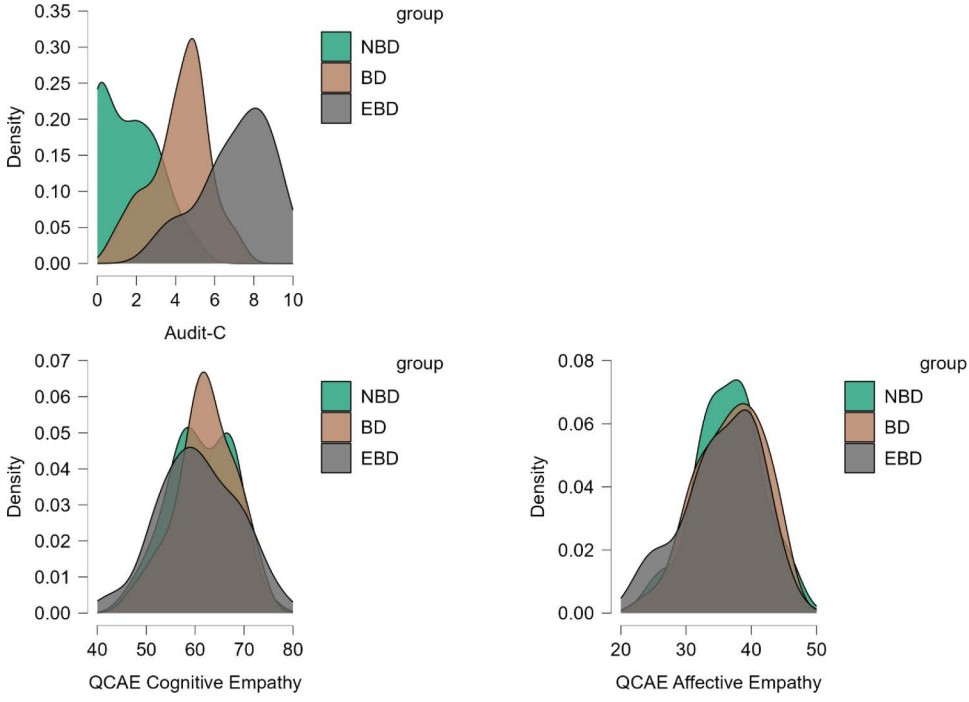

**Fig 1. AUDIT-C, QCAE Cognitive Empathy, and QCAE Affective Empathy scores.** AUDIT-C [40], QCAE Cognitive Empathy, and QCAE Affective Empathy [44] scores, separated by drinking group (non-binge drinking [NBD], binge drinking [BD], and extreme binge drinking [EBD]).

To examine the relative contributions of cognitive and affective empathy to drinking group membership, we conducted a multinomial logistic regression analysis, controlling for age and gender. The overall model was not significant ($p = .24$).

Similarly, one-way ANOVAs with drinking group (non-BD, BD, and EBD) as the factor revealed no significant group differences in cognitive empathy ($p = .68$) or affective empathy ($p = .63$).

Finally, a regression analysis indicated that neither cognitive ($p = .78$) nor affective empathy ($p = .33$) significantly predicted AUDIT-C scores when controlling for age ($p = .03$) and gender ($p = .002$).

## Discussion

The present preregistered study aimed to investigate the relationship between extreme drinking behavior and empathy in a sample of late-adolescent U.S. college students. We hypothesized a dissociation between cognitive and affective empathy, such that affective empathy would be negatively associated with hazardous consumption, whereas cognitive empathy would show no significant association. Despite adequate statistical power to detect medium-to-large-sized effects and contrary to our hypotheses and prior research [24,35], correlation and regression analyses revealed no significant effects for affective empathy: it was neither related to extreme drinking behavior (drinking group) nor hazardous drinking (continuous assessment) in this sample. As predicted, cognitive empathy was not associated with alcohol misuse.

Differences in how empathy is defined [30] and, consequently, in the measures used across studies, which vary in reliability and dimensionality, may account for discrepancies between our findings and previous research. For example, Maurage et al. [35] and Laghi et al. [24] assessed empathy in relation to binge drinking with instruments other than the QCAE [44], such as the Interpersonal Reactivity Index (IRI) [54], Empathy Quotient [55] and the Basic Empathy Scale (BES) [56]. Although these measures index empathic processes, they are not isomorphic (e.g., Michaels et al. [57] reported only moderate correlations between the IRI and QCAE). In addition, the BES demonstrates considerable variability in internal

consistency across studies (ranging from low to high; for review, [58]) and has been critiqued for its unidimensional structure, which limits its ability to differentiate between cognitive and affective components of empathy [44]. The present study employed the multifaceted QCAE [44], a psychometrically validated and reliable instrument, which both differentiates the current design from prior work and may partly account for the failure to replicate previous findings.

In line with prior research, our findings revealed that men were more likely than women to belong to the EBD group [(9, see 59 for review)]. This gender difference in binge drinking has been observed across cultures and countries; however, the size of the gap varies, reflecting the influence of both biological differences and culturally shaped gender roles [(for review, 59)]. Further, methodological differences, such as using gender-specific thresholds for heavy drinking, can affect the observed patterns but do not appear to eliminate the overall gender disparity [(for review, 59)]. Importantly, these findings underscore the need for future research to explore gender-specific motivations for extreme alcohol consumption across cultural contexts and over time, particularly given emerging evidence that this pattern may be reversing, with women increasingly represented among extreme binge drinkers [([8],see [60]: between 2021–2023, women aged 18–25 exceeded men in rates of binge drinking)].

Our findings also highlight the prevalence of EBD in college-aged populations. Data from a U.S. national survey indicate that 12% of college students, and an equal proportion of their non-college peers, reported consuming 10 + drinks in a row (EBD) in 2020 ([11], Table 8-4), which is notably lower than the 20.69% observed in our sample. The higher rate of EBD in our sample suggests that contextual factors such as local drinking culture or university-specific norms may be contributing to alcohol consumption, particularly given that our convenience sample was drawn from two specific universities. This finding is noteworthy because it challenges expectations informed by national averages and highlights the potential influence of contextual and environmental factors on alcohol use among college students.

## Limitations

Several limitations must be acknowledged. First, the cross-sectional design precludes causal inference; for example, it remains unclear whether the observed positive correlation between hazardous drinking and age indicates increased risk with advancing age. Future research should also employ longitudinal designs to examine whether–and under what conditions– both stable (trait) and situational (state) aspects of empathy relate to the development and progression of binge drinking [e.g., 39].

Second, the sample comprised exclusively college students, who may experience unique pressures and motivations for drinking, which limits the generalizability of the findings beyond college populations. Relative to their non-college peers, college students' alcohol use is more strongly influenced by perceived peer norms, suggesting that the college environment may amplify the impact of social influences on drinking behavior [61]. College students also represent a unique population in which to study empathy, as prior research using the IRI has documented gradual declines in empathy across cohorts, with more recent generations scoring lower on the subscales Empathic Concern and Perspective Taking [62, for a meta-analysis].

Further, because participation was voluntary, self-selection bias may have inflated the observed prevalence of alcohol use in this sample, as students with more regular or heavier drinking patterns may have been more likely to participate. Lastly, reliance on self-report measures for both alcohol consumption and empathy may have introduced bias, such as underreporting of drinking behaviors or socially desirable responding on empathy items. As it pertains to alcohol consumption, college students often misjudge standard drink volumes and tend to overestimate the appropriate amounts, leading them to overpour drinks and consequently underreport their actual drinking levels [63]. This raises the possibility that rates at or above BD thresholds were underreported in the present investigation [(see also 64)]. Taken together, the reliance on self-report measures, a convenience sample drawn from only two universities, and the voluntary nature of participation raise the possibility that the nonsignificant findings may reflect characteristics of this sample rather than a true absence of association. Accordingly, the results should be interpreted with caution and warrant further investigation.

## Future directions

Future research should seek to replicate these findings across diverse populations and age cohorts using probability-based sampling, ensure balanced gender representation, and include non-college comparison groups. Additional variables of interest that may influence alcohol use include self-efficacy in resisting peer pressure [e.g., 24], popularity, and familial alcohol consumption. Previous research suggests that adolescents with a family history of alcohol misuse are more likely to exhibit alcohol-related problems [65, 66, 67, see 7 for review]. However, more recent work has begun to question the strength of this association, with Tschorn et al. [68] proposing that familial risk factors may be less influential than previously thought. Instead, social and personality factors, such as impulsivity and risk-taking, appear to play a more substantial role in the development of alcohol misuse. In addition to these influences, prior work has linked attention deficit hyperactivity disorder (ADHD) to binge drinking among college students [69]. Notably, adults with subclinical ADHD have been found to exhibit impairments in affective, but not cognitive, empathy [70], highlighting a potential pathway linking empathy-related processes to problematic drinking.

Looking ahead, future studies may benefit from incorporating experimental paradigms to assess empathy (e.g., [71,72]), as evidence from experimental pain research suggests an additional mechanism linking alcohol use and pain empathy. Alcohol exerts reliable analgesic effects [for a meta-analysis, 73], with animal models indicating that these effects are mediated through opioid-related pathways [74]. Importantly, studies manipulating pain perception through placebo-analgesia show that reducing one's own pain diminishes empathic responses to others' pain [75, see 76 for review]. Similarly, acetaminophen (paracetamol) has been found to blunt empathic responses to both others' physical and social pain [77]. Extending this line of research, it is important to examine whether alcohol's opioid-related analgesic properties differentially impact empathic responding – for example, by diminishing empathy for pain more strongly than for other domains, such as empathy for unpleasant touch. This idea is informed by Rütgen et al. [78], who found that placebo analgesia reduced empathy for both pain and unpleasant touch, indicating that the effect extends, at least in part, to unpleasant affect; however, only pain empathy was blocked by an opioid antagonist, providing evidence that opioid-related mechanisms specifically underlie empathy for pain. Consequently, research on alcohol's social effects should distinguish between empathy domains and explore the possibility that alcohol and pharmacological analgesics act through overlapping neurochemical pathways to influence empathic behavior.

Finally, manipulations of pain perception also have real-life consequences. Hartmann et al. [79] showed that placebo analgesia not only reduces empathy for pain but also diminishes prosocial behavior, such as the willingness to exert effort to help. Such findings raise the possibility of a vicious cycle in which alcohol consumption dampens empathy and prosocial behavior, thereby reducing reciprocal social support and ultimately perpetuating a spiral that exacerbates alcohol-related problems. This reinforces the broader societal implications of alcohol's effects on empathy, and the need to consider both neural mechanisms and social dynamics when addressing alcohol misuse.

## Conclusion

Despite its limitations, this study makes several important contributions. It is among the few to distinguish between extreme binge drinking (EBD) and standard binge drinking (BD) when examining the roles of cognitive and affective empathy, offering a more nuanced approach to understanding alcohol use behaviors. It further reveals rates of alcohol consumption that exceed national averages for late adolescents, underscoring the importance of contextual and population-specific factors. These findings make clear that progress in this field depends on distinguishing between extreme drinking patterns, situating them within their social and cultural contexts, and adopting standardized, psychometrically robust measures of empathy.

## Supporting information

**S1 File. Inclusivity in Global Research Questionnaire.**
(PDF)

## Acknowledgments

 This work was made possible through participation in the IES Abroad Vienna internship program. We thank Anja Gaiswinkler, Anna Kulesza and Emma Wood for proofreading the final version of the manuscript.

## Author contributions

**Conceptualization:** Brooke A. Lester, Nita Bislimi, Claus Lamm, Ekaterina Pronizius.

**Data curation:** Brooke A. Lester, Ekaterina Pronizius.

**Formal analysis:** Brooke A. Lester, Ekaterina Pronizius.

**Funding acquisition:** Claus Lamm.

**Investigation:** Brooke A. Lester.

**Methodology:** Brooke A. Lester, Ekaterina Pronizius.

**Project administration:** Brooke A. Lester, Ekaterina Pronizius.

**Resources:** Claus Lamm.

**Software:** Brooke A. Lester.

**Supervision:** Claus Lamm, Ekaterina Pronizius.

**Validation:** Brooke A. Lester, Nita Bislimi, Ekaterina Pronizius.

**Visualization:** Brooke A. Lester, Ekaterina Pronizius.

**Writing – original draft:** Brooke A. Lester, Nita Bislimi, Ekaterina Pronizius.

**Writing – review & editing:** Brooke A. Lester, Nita Bislimi, Claus Lamm, Ekaterina Pronizius.

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
