## [Decision Letter · Decision Letter 0]

9 Sep 2024

Dear Dr. Lamm,

Thank you for submitting your manuscript to PLOS ONE. After careful consideration, we feel that it has merit but does not fully meet PLOS ONE’s publication criteria as it currently stands. Therefore, we invite you to submit a revised version of the manuscript that addresses the points raised during the review process.

We look forward to receiving your revised manuscript.

Kind regards,

Leona Cilar Budler

Academic Editor

PLOS ONE

**Journal Requirements:**

3. We notice that your supplementary tables are included in the manuscript file. Please remove them and upload them with the file type 'Supporting Information'. Please ensure that each Supporting Information file has a legend listed in the manuscript after the references list.

**Additional Editor Comments:**

There are some major issues that reviewers pointed out. In order to improve paper quality authors should read and consider all comments and suggestions.

Reviewers' comments:

Reviewer's Responses to Questions

**Comments to the Author**

1. Is the manuscript technically sound, and do the data support the conclusions?

Reviewer #1: Partly

Reviewer #2: Partly

2. Has the statistical analysis been performed appropriately and rigorously?

Reviewer #1: Yes

Reviewer #2: No

3. Have the authors made all data underlying the findings in their manuscript fully available?

Reviewer #1: Yes

Reviewer #2: No

4. Is the manuscript presented in an intelligible fashion and written in standard English?

Reviewer #1: Yes

Reviewer #2: Yes

**Reviewer #1:**  The manuscript "Cognitive and Affective Empathy as Relating to Binge-Drinking in Late Adolescence" aims to explore the relationship between cognitive and affective empathy and binge-like consumption among college students in the United States. Although the study presents a relevant issue, some points that negatively affect the results and conclusion.

Minor concerns: the authors should standardize the terms when the abbreviation was admitted (i.e., binge drinking or BD, extreme binge drinking or EBD, etc).

Major concerns:

On page 3, the binge drinking is 0.08g percent or 0.08g/dL, not only 0.08g.

In the definition of adolescence age, I suggest including the WHO recommendations.

On page 5, the authors affirm “Literature indicates that alcohol exposure is not associated with a general reduction in empathy overall, but rather, a dissociated pattern of empathy characterized by impaired affective empathy and preserved cognitive empathy (Maurage et al, 2011), in alcoholic adults. This pattern of preserved affective empathy and impaired cognitive empathy was also observed in adolescents

(Laghi et al., 2019).”. Are these sentences correct?

Several terms have been modified to reduce stigma and negative bias in addiction (see https://nida.nih.gov/nidamed-medical-health-professionals/health-professions-education/words-matter-terms-to-use-avoid-when-talking-about-addiction). Thus, I suggest modifying the terms “abuse”, “drinking habits”, etc.

It is not clear the difference between extreme binge drinking vs problematic drinking.

Family drinking consumption is an important neglected variable, which may underlie adolescent behavior and consumption.

Why male participants were more likely to be in the EBD group than females?

In my opinion, the study was not representative of the USA, since only two universities were explored. The question that remains is whether the results are cultural characteristics of Virginia state and reflect American adolescents.

**Reviewer #2:**  Thank you for the opportunity to review this study. Attached are a number of critiques I hope will improve the study. Most focus on methods, controls, consideration of patterned responses in online survey data, sample demographics and method of sampling, as well as consideration of limitations that may temper results. I also include a few articles that may help with interpretation of your data to provide a more holistic introduction to the study to more adequately represent the current science on this topic.

**Do you want your identity to be public for this peer review?** For information about this choice, including consent withdrawal, please see our Privacy Policy

Reviewer #1: No

Reviewer #2: No

---

## [Decision Letter · Decision Letter 1]

4 Aug 2025

Dear Dr. Lamm,

Thank you for submitting your manuscript to PLOS ONE. After careful consideration, we feel that it has merit but does not fully meet PLOS ONE’s publication criteria as it currently stands. Therefore, we invite you to submit a revised version of the manuscript that addresses the points raised during the review process.

We look forward to receiving your revised manuscript.

Kind regards,

Leona Cilar Budler

Academic Editor

PLOS ONE

Journal Requirements:

Additional Editor Comments (if provided):

There are some issues listed by the reviewer. Please read carefully all comments and suggestions to improve paper quality. Reviewer pointed out that the study justification is not clear. Also, check all journal guidelines (reporting results, citation, etc.).

Reviewers' comments:

Reviewer's Responses to Questions

**Comments to the Author**

Reviewer #1: All comments have been addressed

Reviewer #3: (No Response)

2. Is the manuscript technically sound, and do the data support the conclusions?

Reviewer #1: Yes

Reviewer #3: Partly

3. Has the statistical analysis been performed appropriately and rigorously?

Reviewer #1: Yes

Reviewer #3: I Don't Know

4. Have the authors made all data underlying the findings in their manuscript fully available?

Reviewer #1: Yes

Reviewer #3: Yes

5. Is the manuscript presented in an intelligible fashion and written in standard English?

Reviewer #1: Yes

Reviewer #3: No

Reviewer #1: (No Response)

Reviewer #3: I commend the authors for the time and effort they put in to address previous comments from reviewers.

Introduction:

1. The introduction is disjointed. The paragraph describing the authors’ definition of “adolescence” seems out place and interrupts the flow of their argument.

2. Additionally, the authors restate their study aims multiple times in the introduction (lines 31-32; 52-58; 123-128). I recommend they restructure the section and remove any redundancy.

3. Some of the methods are presented in the introduction (lines 55-58).

Methods:

1. Line 132 says they targeted students attending a university but in line 157-158, they name two universities. Moreover, what was the rationale for selecting these institutions?

2. Line 132-133: need consistent formatting of dates

3. Some of the study’s results are reported in the methods sections, i.e. the demographic breakdown of the final sample and the number of participants excluded from the final analysis. I recommend authors move the information to the results section and/or expanding table 1 to also include the gender and racial breakdowns of the final sample

Technical comments:

1. I recommend the authors to do a thorough copyedit of the manuscript.

2. There were a few inconsistencies with abbreviations. For example: lines 32, 56

3. There are inconsistencies with citation styles within the paper.

**Do you want your identity to be public for this peer review?** For information about this choice, including consent withdrawal, please see our Privacy Policy

Reviewer #1: No

Reviewer #3: No

---

## [Decision Letter · Decision Letter 2]

13 Jan 2026

Cognitive and Affective Empathy in Binge-Drinking during Late Adolescence

PONE-D-24-23331R2

Dear Dr. Pronizius,

We’re pleased to inform you that your manuscript has been judged scientifically suitable for publication and will be formally accepted for publication once it meets all outstanding technical requirements.

Kind regards,

Leona Cilar Budler

Academic Editor

PLOS One

Additional Editor Comments (optional):/

Reviewers' comments:/

Reviewer's Responses to Questions

**Comments to the Author**

Reviewer #1: (No Response)

2. Is the manuscript technically sound, and do the data support the conclusions?

Reviewer #1: (No Response)

3. Has the statistical analysis been performed appropriately and rigorously?

Reviewer #1: (No Response)

4. Have the authors made all data underlying the findings in their manuscript fully available?

Reviewer #1: (No Response)

5. Is the manuscript presented in an intelligible fashion and written in standard English?

Reviewer #1: (No Response)

Reviewer #1: (No Response)

**Do you want your identity to be public for this peer review?** For information about this choice, including consent withdrawal, please see our Privacy Policy

Reviewer #1: No

---

## [Editor Report · Acceptance letter]

PONE-D-24-23331R2

PLOS One

Dear Dr. Pronizius,

I'm pleased to inform you that your manuscript has been deemed suitable for publication in PLOS One. Congratulations! Your manuscript is now being handed over to our production team.

Kind regards,

on behalf of

Dr. Leona Cilar Budler

%CORR_ED_EDITOR_ROLE%

PLOS One